# Contribution of Whole-Genome Sequencing and Transcript Analysis to Decipher Retinal Diseases Associated with *MFSD8* Variants

**DOI:** 10.3390/ijms23084294

**Published:** 2022-04-13

**Authors:** Anaïs F. Poncet, Olivier Grunewald, Veronika Vaclavik, Isabelle Meunier, Isabelle Drumare, Valérie Pelletier, Béatrice Bocquet, Margarita G. Todorova, Anne-Gaëlle Le Moing, Aurore Devos, Daniel F. Schorderet, Florence Jobic, Sabine Defoort-Dhellemmes, Hélène Dollfus, Vasily M. Smirnov, Claire-Marie Dhaenens

**Affiliations:** 1Univ. Lille, Inserm, CHU Lille, U1172-LilNCog-Lille Neuroscience & Cognition, F-59000 Lille, France; anais.poncet@chu-lille.fr (A.F.P.); olivier.grunewald@chru-lille.fr (O.G.); aurore.devos@chru-lille.fr (A.D.); 2University of Lausanne, Jules-Gonin Eye Hospital, 1004 Lausanne, Switzerland; veronika.vaclavik@fa2.ch; 3Cantonal Hospital, Department of Ophthalmology, 1700 Fribourg, Switzerland; 4National Reference Centre for Inherited Sensory Diseases, University of Montpellier, Montpellier University Hospital, Sensgene Care Network, ERN-EYE Network, F-34000 Montpellier, France; i-meunier@chu-montpellier.fr (I.M.); beatrice.bocquet@inserm.fr (B.B.); 5Institute for Neurosciences of Montpellier (INM), University of Montpellier, INSERM, F-34000 Montpellier, France; 6Exploration de la Vision et Neuro-Ophtalmology, CHU de Lille, F-59000 Lille, France; isabelle.drumare@chru-lille.fr (I.D.); sabine.defoort@chru-lille.fr (S.D.-D.); vasily.smirnov@chru-lille.fr (V.M.S.); 7Centre de Référence pour les Affections Rares en Génétique Ophtalmologiques, Hopitaux Universitaires de Strasbourg, F-67000 Strasbourg, France; valerie.pelletier@chru-strasbourg.fr (V.P.); dollfus@unistra.fr (H.D.); 8Department of Ophthalmology, Cantonal Hospital, 9007 St. Gallen, Switzerland; margarita.todorova@kssg.ch; 9Department of Ophthalmology, University of Zürich, 8091 Zürich, Switzerland; 10Department of Ophthalmology, University of Basel, 4056 Basel, Switzerland; 11Department of Child Neurology, Amiens-Picardy University Hospital, F-80000 Amiens, France; LeMoing.Anne-Gaelle@chu-amiens.fr; 12Faculty of Biology and Medicine, University of Lausanne and Faculty of Life Sciences, Ecole Polytechnique Fédérale of Lausanne, 1004 Lausanne, Switzerland; daniel.schorderet@hin.ch; 13Unité de Génétique Médicale et Oncogénétique, Centre Hospitalier Universitaire Amiens Picardie, F-80000 Amiens, France; Jobic.Florence@chu-amiens.fr; 14Université de Lille, Faculté de Médecine, F-59000 Lille, France

**Keywords:** *MFSD8* gene, isolated macular dystrophy, neuronal ceroid lipofuscinosis, deep intronic variant, transcript analysis

## Abstract

Biallelic gene defects in *MFSD8* are not only a cause of the late-infantile form of neuronal ceroid lipofuscinosis, but also of rare isolated retinal degeneration. We report clinical and genetic data of seven patients compound heterozygous or homozygous for variants in *MFSD8*, issued from a French cohort with inherited retinal degeneration, and two additional patients retrieved from a Swiss cohort. Next-generation sequencing of large panels combined with whole-genome sequencing allowed for the identification of twelve variants from which seven were novel. Among them were one deep intronic variant c.998+1669A>G, one large deletion encompassing exon 9 and 10, and a silent change c.750A>G. Transcript analysis performed on patients’ lymphoblastoid cell lines revealed the creation of a donor splice site by c.998+1669A>G, resulting in a 140 bp pseudoexon insertion in intron 10. Variant c.750A>G produced exon 8 skipping. In silico and in cellulo studies of these variants allowed us to assign the pathogenic effect, and showed that the combination of at least one severe variant with a moderate one leads to isolated retinal dystrophy, whereas the combination in trans of two severe variants is responsible for early onset severe retinal dystrophy in the context of late-infantile neuronal ceroid lipofuscinosis.

## 1. Introduction

Inherited retinal degenerations (IRD) affect about two million people worldwide and share a common feature of progressive degeneration of photoreceptors and/or retinal pigment epithelium. Three sub-types of IRD—macular dystrophy (MD), cone dystrophy (COD) and cone–rod dystrophy (CORD)—manifest in primary loss of central vision, photophobia and colour vision disturbances. Inherited MD first affects the central zone of the retina. However, some MD cases evolve into a more widespread retinal disease at the late stages due to variable cone or cone–rod dysfunction [1]. The archetypal MD is Stargardt disease (STGD1, OMIM # 248200). COD and CORD constitute a heterogeneous group of disorders primarily affecting cone photoreceptors [2,3]. COD differs from CORD by the absence of night blindness, which occurs in the latter due to concomitant rod degeneration. However, most of the patients with COD develop rod degeneration later in the progression of disease [3,4]. 

Since the introduction of next-generation sequencing in the field of IRD, a growing number of variants have been identified in genes known to be associated with syndromic disorders, despite a clinical presentation restricted to the retina. Isolated retinal degeneration can thus be caused by variants in *USH2A* [5,6,7], *BBS1* [8], or more recently, *CLN3* [9,10] and *CLN5* [11]. Lately, *MFSD8* was incriminated in autosomal recessive MD, COD or CORD [12,13,14,15,16,17]. *MFSD8* (major facilitator superfamily domain-containing 8) (MIM * 611124, NM_152778.3) was originally associated with neuronal ceroid lipofuscinosis (NCLs), a group of severe neurodegenerative disorders, characterized by the lysosomal accumulation of abnormal autofluorescent material (lipofuscin-like ceroid lipopigments) leading to selective damage and loss of neurons [18,19]. Pathogenic variants in *MFSD8* cause a variant late-infantile form of NCL (vLINCL) called CLN7 disease (MIM # 610951) [20]. The clinical course includes psychomotor decline, seizures, visual failure and reduced lifespan [21]. *MFSD8* encodes a CLN7 protein localized to the lysosomal membrane [22,23,24]. Its exact function remains unclear, but roles in lysosome trafficking [25,26] have been supported.

In this study, we report ocular phenotype of patients with *MFSD8*-related isolated retinal dystrophy and LINCL, describe novel *MFSD8* variants and their functional repercussion, and present a tentative of genotype–phenotype correlation. 

## 2. Results

### 2.1. Patients Description

*MFSD8* variants have been identified in six unrelated patients with isolated retinal dystrophy and in one patient with late-infantile neuronal ceroid lipofuscinosis. Clinical data of patients on initial examination are summarized in Table 1 and follow-up data for the five oldest patients in Appendix A. At first examination, two patients had a clinical picture of MD, three patients had COD, one patient presented CORD and one patient early onset severe retinal dystrophy (EOSRD).

Three patients (L-08031428, VV-1595021, IM-190703) presented with teenage-onset (age 12, 12 and 14, respectively) MD, COD or CORD. First complaints were progressive visual loss, poor colour discrimination and photophobia. When realized, visual fields depicted relative central scotoma. Colour vision tests were variably affected (from normal colour perception to without-axis severe errors on ranking tests). ffERG was normal for one patient (VV-1595021: MD). Light-adapted responses only were reduced in one case (IM-190703: COD). One patient depicted a cone–rod pattern of retinal dysfunction (L-08031428: CORD, Appendix A). Fundus examination revealed an optic disc pallor and a punched-out atrophic round macular lesion with slightly hyperpigmented borders (Figure 1, teenage-onset forms). Foveal lesion appeared dark with hyperautofluorescent edges on SW-FAF. SD-OCT found either an aspect of foveal cavitation in outer reflective layers (L-08031428, VV-1595021) or a larger destruction of outer reflective layers (IM-190703) in the macula. Long-term follow-up data were available for two patients (L-08031428, VV-1595021). BCVA gradually worsened. Patient L-08031428 became legally blind at 25 years old. Light-adapted ffERG responses became undetectable at 28 years. ffERG was not repeated for VV-1595021. Initial foveal lesion progressed to a bull’s eye maculopathy in both patients and then to mid-peripheral retinal depigmentation with bone spicule-like pigment migration in L-08031428. SD-OCT aspect of foveal cavitation collapsed by 20 years in both patients and then progressed to a widespread destruction of outer reflective layers in L-08031428. 

Three patients (HD-OPH1206, HD-OPH4200, VV-51717) presented adult-onset (age 30, 37 and 55, respectively) maculopathy or COD. Retinal picture was quite similar. Patients complained of progressive BCVA loss (mostly reading difficulties) and photophobia. Visual field discovered central scotoma. ffERG was normal (VV-51717, MD, Appendix A) or depicted a reduction in light-adapted responses (HD-OPH1206 and HD-OPH4200: COD). Fundus depicted bull’s eye maculopathy (Figure 1, adult-onset forms). It was a dark fovea with hyperautofluorescent edges on SW-FAF corresponding to a more or less large interruption of outer reflective layers on SD-OCT. On follow-up, patients depicted slow progression of macular lesions. 

One patient presenting LINCL was also identified (L-20021807). Visual acuity loss was first reported at the age of 6. He complained of mild photophobia. Myoclonic seizures appeared at 8 and required polymedication (levetiracetam, carbamazepine). Despite this therapy, seizures were only partially controlled. Brain MRI discovered upper vermis atrophy. At the age of 9, parents noted behavioural changes: attention deficit and aggressivity. This patient was referred to our department at 10 years old. BCVA was limited to 20/400 OU; patient had eccentric fixation with “overlooking”. Kinetic perimetry disclosed a 20° relative central scotoma at III3e target. Peripheral isopter at V4e target was full. ffERG was unrecordable (Appendix A). Fundus examination revealed a waxy pallor of the optic disc, severe vascular attenuation and macular depigmentation. There was a cellophane light reflex and yellowish discoloration of the macula (Figure 2). SW-FAF showed a large area of increased autofluorescence with indistinct border in the posterior pole and the second more narrow ring of increased autofluorescence around the fovea. Both foveae were irregularly hypoautofluorescent. The peripheral retina was isoautofluorescent. On SD-OCT, the retina was thin with indistinct lamination and disappearance of outer reflective layers. 

### 2.2. Genotyping and Transcript Analysis

Variants in the *MFSD8* gene have been identified in five patients in a cohort of 1049 IRD cases analysed by next-generation sequencing (NGS) of large panels, among which 340 presented MD, COD or CORD (Appendix A). Two additional patients from a Swiss cohort were added to this group. Molecular analysis revealed the presence of twelve different variants in *MFSD8*, among which seven were novel: four missense, one nonsense, one deep intronic variant and one large deletion encompassing two exons. A silent change found twice in our cohort was reported by Reith and colleagues during the submission of this manuscript [27].

All six patients with isolated retinal dystrophy carried at least one missense variant, the second hit being either another missense or a loss-of-function variant (Table 2). Among the novel missense variants, the c.1006G>A, p.(Glu336Lys) variant has never been reported before, while the c.1006G>C p.(Glu336Gln) was previously described in MD [12,13] and identified in two patients with MD in this series. Residue Glu336 is located in the extracellular loop L9 (Figure 3). Variants c.104G>A p.(Arg35Gln) and c.155G>C p.(Gly52Ala) are associated in a complex allele. The pathogenic variant of this allele could be p.(Gly52Ala), classified as possibly pathogenic variant class 4, while p.(Arg35Gln) is predicted to be a variant of unknown significance according to ACMG 2015 criteria. The novel p.(Arg337Cys) variant is rare and presents a high CADD score. 

Because only one variant was found in the *MFSD8* gene by NGS in patient HD-OPH1206, we completed the genetic study by whole-genome sequencing. Intron analysis enabled the identification of a second anomaly in trans: the novel deep intronic variant c.998+1669A>G (Appendix A). This variant was predicted to create a splice donor site (SpliceAI donor gain score 0.68) (Table 2), with an acceptor site 140 bp upstream of this position (SpliceAI acceptor gain score 0.30). The transcript study in patient LCLs confirmed the insertion of a 140 bp pseudoexon between exons 10 and 11 (Figure 4a). The semi-quantification showed 77% of residual normal transcript, corresponding to the sum of the wild-type and the mutant alleles. The specific fraction due to the mutant c.998+1669A>G allele can thus be estimated at around 50%.

One patient with MD (L-08031428, 12 years old) and one with LINCL (L-20021807, 10 years old), both native from the North of France, carried the same variant c.750A>G, not reported in gnomAD. This presumably silent change variant was located five nucleotides upstream of the 3′ end of exon 8, and the SpliceAI splice prediction tool suggested a splice donor site loss (score 0.45). Transcript analysis of lymphoblastoid cell lines of both patients revealed the presence of one normal transcript and another shorter one lacking exon 8, in accordance with in silico prediction (Table 2, Figure 4b). The exon 8 loss is an out-of-frame deletion, and therefore leads to a null allele. The higher expression of the mutant in the presence of puromycin confirmed the transcript degradation by the nonsense-mediated decay (NMD) pathway. The transcript semi-quantification differed between the two families: before puromycin treatment, 25% of the normal transcript remained for patient L-20021807 (LINCL), and 60% for patient L-08031428 (MD). Of note, in LCLs, the two *MFSD8* alleles cannot be separated and are both part of the semi-quantification. The remaining normal transcript amount for the LINCL case reflects the specific effect of the mutant c.750A>G allele, as the other allele carrying the large exons 9–10 deletion undergoes NMD. On the contrary, 60% of the residual normal transcript in MD patient corresponds to the sum of the wild-type and mutant *MFSD8* alleles. 

In patient L-20021807, the second variant is a large deletion of exons 9 and 10, c.(755-2726_998+1981)delinsGTA (Appendix A). It was first detected by an NGS panel and then confirmed by whole-genome sequencing which defined the exact borders. This 7118 bp deletion leads also to a frameshift and therefore to a null allele. Repetitive elements were investigated at the breakpoints to decipher the causal mechanism. DNA elements and one LINE sequence have been identified at the 5′ and 3′ breakpoints, respectively.

## 3. Discussion

This study of seven patients carrying *MFSD8* variants showed that isolated retinal diseases are associated with combination of one severe and one mild or moderately severe variant, whereas the syndromic form with an EOSRD phenotype seems to be related to severe variants. Our results confirm that *MFSD8* is a rare cause of retinal degeneration as patients carrying biallelic variants in this gene represent only 5/1049 (0.47%) of the total number of cases sequenced by large NGS panels and 5/340 (1.5%) of the patients with macular dystrophy or cone/cone–rod dystrophy. This is very different from *ABCA4,* the most prevalent gene related to autosomal recessive MD and COD/CORD (597/1293, 46% in our cohort). However, *MFSD8* should be included in NGS strategies when investigating patients with IRD. It is very likely that some of our unsolved patients analysed by a small, targeted gene panel in the past (*n* = 356, Appendix A) are carriers of *MFSD8* variants. A complementary study would allow us to address this point. 

### 3.1. Common or Specific Clinical Features 

Patients with isolated retinal dystrophy linked to *MFSD8* presented MD, COD or CORD. In two patients (L-08031428 and VV-1595021), a foveal cavitation was an early OCT feature before the collapse of inner reflective layers occurring by their twenties. Two group of age at onset were clearly distinct: early teenage onset (from 12 to 14 years in our patients) and adulthood onset (from 30 to 55 years).

The *MFSD8*-linked macular lesions mimic Stargardt disease and *ABCA4*-related COD and CORD: (i) two similar peaks of onset ages: late childhood [32,33] and late adulthood [34,35]; (ii) a typical foveal cavitation progressing to a collapse of outer retina and then to a bull’s eye macular lesion in early onset forms [36,37]; (iii) a subset of STGD1 patients had a cone or cone–rod pattern of retinal dysfunction at ffERG [38,39], as seen in patients in this series. However, macular lesions in our *MFSD8* patients were grossly roundish (except for HD-OPH1206), while they are usually larger in STGD1. No patient in our series developed satellite hyperautofluorescent flecks, typical in STGD1 [40], and no peripapillary retinal sparing [41,42] was observed. 

Clinical presentation of patient L-20021807 with LINCL was more precocious and severe: he was legally blind before the age of 10. Retinal degeneration with unrecordable ffERG and a widespread outer reflective layer destruction on SD-OCT was a feature at first examination. There was no similarity with STGD1, neither in retinal function (unrecordable ffERG responses), nor in fundus (waxy pallor of optic disc, vascular attenuation and posterior pole depigmentation) or in multimodal imaging appearance (ring of increased autofluorescence on SWAF and widespread outer retinal layer destruction on HD-OCT). This clinical phenotype is in keeping with EOSRD [43,44] and is reminiscent of other types of NCL [45,46,47]. 

### 3.2. Novel Splice Defects

We report here the first deep intronic variant in the *MFSD8* gene, leading to the creation of a splice donor site in intron 10, and producing a 140 bp pseudoexon insertion. Introns of *MFSD8* have not yet been fully investigated, and it is still unknown if this variant could be recurrent in other IRD or NCL cases. This discovery underscores the need to complete the full sequencing of *MFSD8* in unsolved cases and in cases where only one pathogenic *MFSD8* variant has been found. Deep intronic splice variants could be good candidates for antisense oligonucleotide therapy, as was previously shown for *MFSD8* [48]. Deep intronic variants in *ABCA4* are also frequently observed in Stargardt disease [49]. 

The second splice defect identified in our patients was c.750A>G, firstly predicted as p.(Glu250=). During the revision in this manuscript, this variant was published at the homozygous state in patients presenting a neuronal ceroid lipofuscinosis [27]. It was previously reported once in the LOVD database as likely benign, due to its supposed absence of effect on the protein. However, it has never been observed in gnomAD nor in our 3348 IRD patients’ cohort, and was found in two patients, both sharing the same geographic origin and carrying a second likely pathogenic variant in *MFSD8* (Table 2). Transcript analysis in immortalized lymphocytic cell lines confirmed exon 8 skipping in the major transcript. Another splice variant, c.754+2T>A has been described by Siintola et al., in 2007 [50], who also showed exon 8 skipping with this variant. In this paper, exon 7 seems also to be alternatively spliced. According to the UCSC database, *MFSD8* contains 10 transcripts differing in their first coding exon and the presence of alternative cassettes: exons 2, 6, 7 and 8 (using NM_152778.3 as a reference). We observed that exclusion of exon 8 is present at very low levels in normal conditions (Figure 4b), but that the c.750A>G variant enhanced the percentage of exclusion in patients. As shown in GTEx portal (https://gtexportal.org/home/gene/MFSD8, 6 March 2022), the alternative exon 8 is differentially expressed in tissues. The presence of the variant could modify the exonic splice regulatory sequence recognized by transcription factors, as predicted by HExoSplice [51]. The exact function of the different *MFSD8* transcripts is still unknown, but a differential expression in tissues [50] might be linked to specific expression in the splice factors repertory and could lead to specific functions such as those reported for other transcripts, such as *CRB1, REEP* and *DFNB31* [52,53,54]. 

### 3.3. Genotype–Phenotype Correlations

All *MFSD8* variants associated with isolated retinal dystrophy (MD, COD/CORD) are located in the transmembrane alpha helices (Figure 3), except p.(Glu381*), located at the most extracellular side of loop 9, nearby the two proteolytic cleavage sites Asn370 and Asn376. Residue Glu336 is located in extracellular loop L9. Electrochemical charge change due to this substitution can destabilize the protein conformation in modulating interactions. However, the Glu-to-Lys substitution is considered moderately conservative as the Grantham score is 56, whereas the Glu-to-Gln change is conservative (Score: 29) [31]. A mild impact is therefore expected for both. Variants c.104G>A p.(Arg35Gln) and c.155G>C p.(Gly52Ala) are associated in a complex allele. The pathogenic variant of this allele could be p.(Gly52Ala), classified as possibly pathogenic variant class 4, while p.(Arg35Gln) is a predicted variant of unknown significance according to ACMG 2015 criteria. They are located in or close to the first transmembrane domain of the protein, p.(Arg35Gln) is on the external cytosolic side of the protein, and interacts with amino acids Ala159, and Glu164 belongs to another alpha helix domain and with Glu27 located at the junction between the first transmembrane domain and the N-terminal extremity (Appendix A). Again, this substitution is conservative (Grantham score: 43). Variant p.(Gly52Ala) is more internal, in an helix encumbered by aromatic residues (Phe53, Tyr83, Phe180). A somewhat higher impact of this change is predicted (Grantham score: 60), but both residues are small and no dramatic destabilization is expected. The p.(Gly52Ala) variant could be the cause for the change in the complex allele, especially since another variation targeting the same residue, p.(Gly52Arg), was already reported as causal in NCL [55], but with a higher impact since it replaces a small residue by a larger and positively charged polar one. The novel p.(Arg337Cys) variant is rare and presents a high CADD score. The substitution Arg into Cys is predicted to have a high impact (Grantham score: 180) as it replaces a long, positively charged residue interacting through hydrogen bonds with Tyr503 and Glu518, located in the last transmembrane domain and in the last amino acid residue, respectively, by a short residue able to form disulphide bridges. On the other hand, the well-known variant p.(Thr294Lys) is a moderately conservative change (Grantham score: 78). Thr294 is located in loop 7, in a very buried zone, encumbered by large aromatic residues Trp300, Tyr298, Phe419 and interacting with Ile290, Thr291 Asn309 by hydrogen bonds. The substitution of a polar, uncharged amino acid residue by a positively charged amino acid residue could modify the protein conformation. Based on 3D structure changes, we observed no substantial destabilization of the protein for most of the novel missense variants reported in our series. We therefore considered them as moderately severe variants (Table 3).

Recurrence was observed for some variants in patients presenting NCL: c.929G>A, p.(Gly310Asp) [50,55] and c.881C>A, p.(Thr294Lys) [24,55,57,58], for which a founder effect has been reported in the population originating from the former Czechoslovakia [56]. Interestingly, in NCL cases, these two variants are in trans with another severe variant (frameshift, splice site variants) or are homozygous, suggesting a severe effect (Table 3). The high pathogenicity of the p.(Thr294Lys) variant was proved by the increase in proteolytic cleavage, leading to a loss of function of the protein [24]. On the contrary, in isolated retinal dystrophy cases from our series, p.(Gly310Asp) and p.(Thr294Lys) are in combination with either a splice variant with partial effect (patient HD-OPH1206) or with p.(Glu336Gln) (patients HD-OPH4200 and VV-51717). The latter is predicted to be a mild allele by in silico analysis and was only reported in MD cases in trans with a loss of function variant [12,13] (Table 2 and Table 3). Regarding the two splice variants investigated here, both produced a partial effect with residual normal transcript, estimated at 25% for c.750A>G and at 50% for c.998+1669A>G, the mutant alleles being degraded by nonsense-mediated decay (NMD) as confirmed by puromycin treatment. The level of the remaining wild-type variant allowed us to consider c.998+1669A>G as a moderately severe variant, and c.750A>G as a severe variant. In MD cases, these two variants are in trans with a missense variant. However, in HD-OPH1206, the missense variant p.(Gly310Asp) is predicted to be the severe variant of the genotype, while in L-08031428, the severe variant should be the c.750A>G (Table 2). In the young patient with LINCL carrying the c.750A>G (L-20021807), the other allele is inactivated by the loss of two exons, leading to a very low level of protein expression. This patient, carrying two severe variants, exhibited therefore a more severe phenotype. The severe effect of this c.750A>G variant was confirmed by the recent description at the homozygous state in two patients from the same family with NCL [27]. As shown in Table 2 and Table 3, all isolated retinal dystrophy cases of our series and from literature harboured one severe and one mild or moderately severe *MFSD8* variant, whereas all LINCL cases carried two severe *MFSD8* variants. The combination in trans of two types of variants, and their impact on protein dysfunction, could therefore explain why a same variant can be found in both early and late phenotypes. The degree of severity is correlated with the residual protein expression in the tissue. The total expression of the CLN7 protein in patients’ cells would be necessary to further confirm this hypothesis. 

## 4. Materials and Methods

### 4.1. Cohort of Patients

Patients harbouring *MFSD8* variants were part of a French cohort including 3348 patients with inherited retinal degeneration (IRD) analysed in our laboratory between 2014 and 2021. Only 1049 patients analysed using a large NGS panel were considered in the following description. This cohort is divided into two groups: (i) a group of 709 patients presenting rod–cone dystrophy, Leber congenital amaurosis, congenital stationary night blindness, or Choroideremia (“Other IRD” in flowchart in Appendix A); (ii) a group of 340 patients presenting cone or cone–rod dystrophy and macular dystrophy including Stargardt disease. Two additional patients were retrieved from an independent Swiss IRD cohort followed at the Jules-Gonin Eye Hospital and analysed in the Institute for Research in Ophthalmology. Those patients were included to strengthen the clinical and genetic description of *MFSD8* variants, rarely reported in isolated macular dystrophy.

### 4.2. Clinical Examination

Prior to testing, written informed consent was obtained from each study participant. The study protocol adhered to the tenets of the Declaration of Helsinki and was approved by the local ethics committees (the list is provided in Institutional Review Board Statement below).

Clinical data were retrospectively collected from medical records. They included sex, age at time of first symptoms, diagnosis and examination, personal or familial history, complaints, best-corrected visual acuity (BCVA), refractive error, slit-lamp biomicroscopy, colour vision (either Ishihara album, Lanthony D-15 panel or Farnsworth 100 Hue) static and kinetic visual fields (VFs), full-field electroretinogram (ffERG according to the standards of International Society for Clinical Electrophysiology of Vision [61]), fundus photography, spectral domain optical coherence tomography (SD-OCT: Spectralis OCT, Heidelberg Engineering, Inc., Heidelberg, Germany) and short-wavelength autofluorescence (SWAF: Heidelberg Retinal Tomograph, Heidelberg Engineering, Inc., Heidelberg, Germany). Only patients presenting MD, COD or CORD were included in the study.

### 4.3. Genetic Assessment

Among the 3348 probands investigated in our laboratory, 1049 were genetically assessed using a large NGS panel of 156 or 230 genes [62] (the latest being an updated version of the first “156 genes” panel) (Appendix A). Both panels contained the *MFSD8* gene. The second part of the cohort, comprising 2299 patients, was analysed using an NGS targeted gene panel, which did not contain the *MFSD8* gene. This group was therefore excluded from the study. Among the 340 patients analysed with the large panel and presenting cone, cone–rod dystrophy and macular dystrophy, five of them harboured variants in *MFSD8* gene. It should be noted that none of the 709 patients with “other IRD” carried a *MFSD8* variant.

Capture oligonucleotide probes were designed using the Haloplex target enrichment System (Agilent Technologies Inc., Santa Clara, CA, USA). DNA libraries were sequenced on a NextSeq500 sequencer (Illumina Inc., San Diego, CA, USA). The data analysis was conducted using an in-house developed pipeline compiling the data obtained from Seqnext (JSI Medical System, Ettenheim, Germany) and GATK software. All single-nucleotide variations were confirmed by bidirectional Sanger sequencing of the *MFSD8* exons and exon–intron boundaries (MIM * 611124, NM_152778.3). PCR and sequencing conditions are available on request (see primers list in Appendix A). Segregation analysis was performed when family members’ DNA samples were available. 

A targeted *MFSD8* complete gene sequencing of patient HD-OPH1206 was performed to search for the second intronic variant (BGI Genomics, Hongkong). High-quality genomic DNA samples were randomly fragmented by Covaris Technology. Fragments of 350 bp were selected. After ligation of adapters, followed by amplification by ligation-mediated PCR (LM-PCR), single-strand separation, and cyclization, DNA fragments were read through on the BGISEQ-500 platform. Sequencing-derived raw image files were processed by BGISEQ-500 base-calling Software (v1) with default parameters, and the sequence data of each individual were generated as paired-end reads. Data of each sample were mapped to the human reference genome (GRCh37/HG19) using Burrows–Wheeler Aligner software (v0.7.17). Calling was performed using Genome Analysis Toolkit. Variant calling was targeted on *MFSD8* intronic variants, filtering on SpliceAI predictions (Score > 0.2). 

### 4.4. Copy Number Variants (CNVs)

The detection of CNVs was performed by quantitative analysis of the data obtained from the bam files. For each patient and for each target (i.e., exon), we calculated the ratio of the depths of the reads for this target/depth of the reads for all targets, and divided this ratio by the full mean coverage for all control samples analysed on the same NGS run. A ratio value of 1 means that copy number is identical to the control samples, and a ratio value of 0.5 reveals only one copy of the allele and a heterozygous deletion. Borders of the CNV identified in patient L-20021807 have been obtained by whole-genome sequencing (BGI Genomics, Hongkong) according to the above protocol. 

### 4.5. Variant Pathogenicity Assessment

Variant pathogenicity was assessed through our in-house pipeline embedding commercially available bioinformatics software and using data available from public variant databases in accordance with the Guidelines of the American College of Medical Genetics and Genomics [28]. 

### 4.6. Lymphoblastoid Cell Lines’ (LCLs) Production

To assess the pathogenicity at the RNA level, of the two splice variants, we requested novel samples for the 3 probands carrying the c.998+1669A>G and c.750G>A variants. Lymphocytes were collected from patient blood samples and EBV (Epstein–Barr Virus) immortalized using a Transfokit-EBV kit provided by the University Hospital of Reims (France). The immortalized lymphocytes were cultured in suspension in RPMI (Gibco™, LifeTechnologies, Carlsbad, CA, USA) supplemented with 15% foetal bovine serum (Gibco™, LifeTechnologies, Carlsbad, CA, USA) and 0.5% Penicillin–Streptomycin (Sigma-Aldrich, St Louis, MO, USA) at 37 °C under 5% CO_2_. After a few weeks, the culture was separated to prepare two cell pellets treated 5h with or without puromycin (Sigma-Aldrich, St Louis, MO, USA), and an NMD inhibitor, at a final concentration of 200 µg/mL.

### 4.7. Transcript Analysis 

RNA extractions from the LCLs pellets were performed with the Nucleospin^®^ RNA kit (Macherey-Nagel, Hoerdt, France) according to the supplier’s protocol. The DNased RNAs obtained were assayed with Nanodrop and the quality was checked using with the Agilent RNA 6000 Nano Chips kit (Agilent Technologies, Santa Clara, CA, USA) on Bioanalyzer 2100. cDNAs were synthesized using the AffinityScript cDNA Synthesis Kit (Agilent Technologies) according to the manufacturer’s recommendations, and regions of interest of cDNAs were amplified using the Taq DNA Polymerase Invitrogen kit (Invitrogen^TM^, Waltham, MA, USA). The PCR products were separated on a 1.5% agarose gel at 120V for 1 h. Quantification of bands on agarose gel were carried out with the Image Lab 6.1 software (Bio-rad). PCR products were purified and sequenced by the Sanger method with the 3730XL DNA Analyzer (Applied Biosystems^TM^, Waltham, MA, USA). The sequences were read using the SeqScape™ software (Applied Biosystems™).

## 5. Conclusions

Isolated retinal dystrophy associated with *MFSD8* gene defects manifests as MD, COD or CORD, and could mimic STGD1. Retinal degeneration in *MFSD8*-linked LINCL presents as EOSRD, and is clearly distinguishable from isolated phenotypes. In silico and in cellulo studies of variants identified in this study and in previously reported patients allowed us to assign a pathogenic effect to variants, and showed that the combination of at least one severe variant with a moderate one leads to isolated retinal dystrophy, whereas the combination in trans of two severe variants is responsible for an EOSRD in the context of late-infantile neuronal ceroid lipofuscinosis.

## Figures and Tables

**Figure 1 ijms-23-04294-f001:**
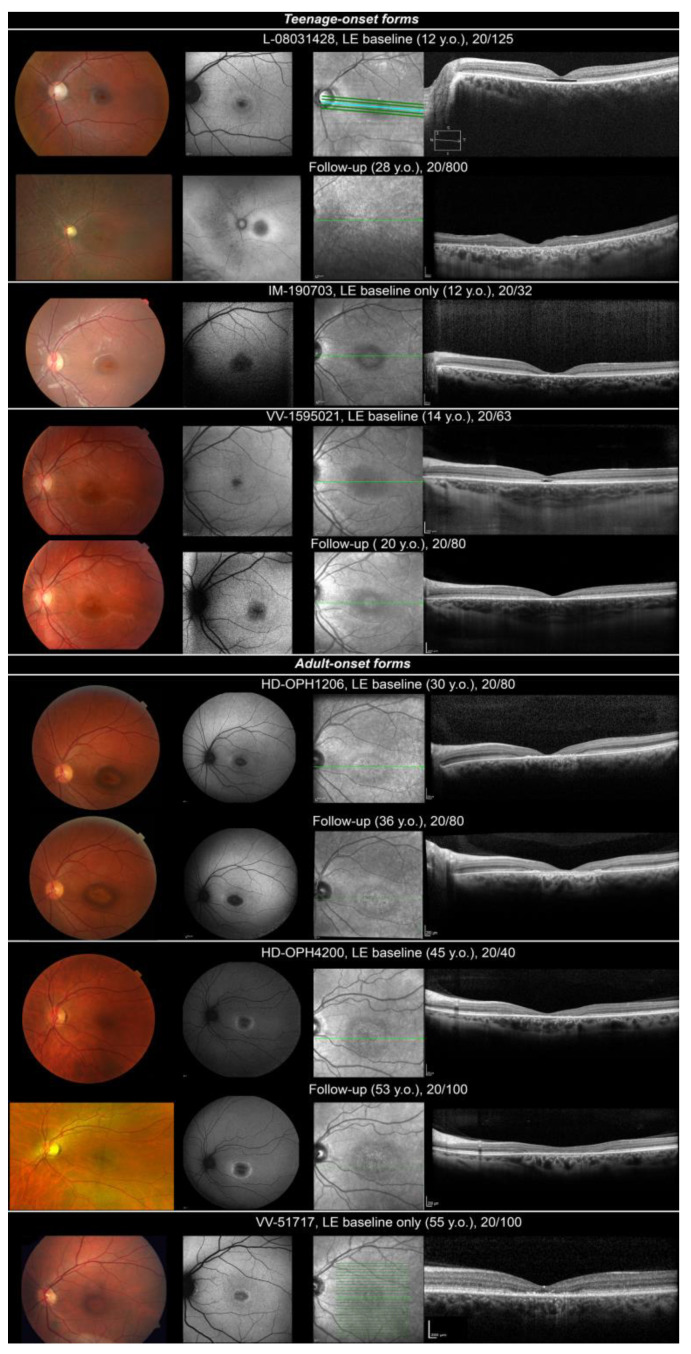
Ocular findings in isolated retinal dystrophy linked to *MFSD8* variants. Fundus photography, SW-FAF, IRR and HD-OCT are shown in successive rows. Top, teenage-onset maculopathy/COD/CORD. Optic disc pallor and a punched-out atrophic foveal lesion with slightly hyperpigmented borders. Foveal lesion looked dark with hyperautofluorescent edges on SW-FAF. SD-OCT found either a temporary aspect of foveal cavitation in outer reflective layers (L-08031428, VV-1595021) or a larger destruction of outer layers (IM-190703) in the macula. Slow progression of macular lesions at follow-up, progression to a more widespread retinal lesions with mid-peripheral involvement in patient L-08031428. Bottom, adult-onset maculopathy/COD. Bull’s eye maculopathy. Dark fovea with hyperautofluorescent edges on SW-FAF corresponding to a more or less large interruption of outer reflective layers on SD-OCT. Slow progression of macular lesion at follow-up.

**Figure 2 ijms-23-04294-f002:**
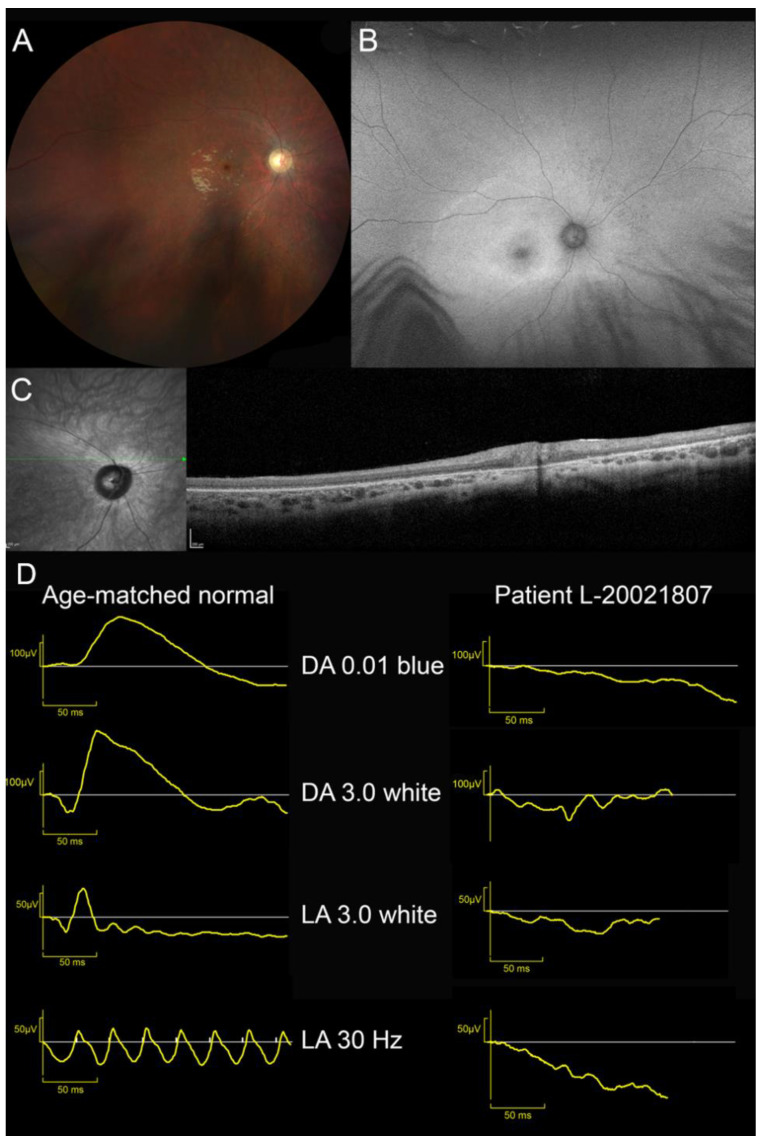
Ocular findings in patient with late-infantile neuronal ceroid lipofuscinosis (MFSD8-LINCL). (**A**) Waxy pallor of optic disc, severe vascular attenuation and whitish, depigmented appearance of posterior pole and midperipheral retina. Cellophane light reflex from the macula. (**B**) SW-FAF showed a large area of increased autofluorescence with indistinct border in the posterior pole and the second more narrow annulus of increased autofluorescence around the fovea. Fovea was irregularly hypoautofluorescent. Peripheral retina was isoautofluorescent. (**C**) SD-OCT. Retina was thin with indistinct lamination and widespread disappearance of outer layers (ONL, EZ and RPE). (**D**) ffERG. Unrecordable responses under dark- and light-adapted conditions.

**Figure 3 ijms-23-04294-f003:**
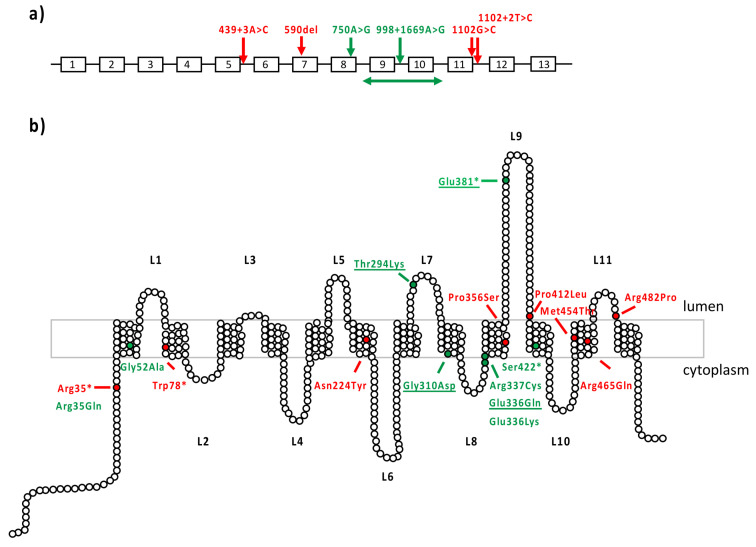
MD-associated *MFSD8* variants. (**a**) Loss-of-function variants reported in macular dystrophy. *MFSD8* gene is represented with its 12 exons. Variants are annotated according to the cDNA nomenclature. In red are variants from the literature and in green are novel variants identified in this study. (**b**) Missense variants reported in macular dystrophy. CLN7 is composed of 12 transmembrane domains and 11 extracellular or cytoplasmic loops. The 518 amino acids of CLN7 protein are depicted as white circles. In red are variants from the literature; in green are novel variants identified in this study; underscored are the already reported variants found in our study.

**Figure 4 ijms-23-04294-f004:**
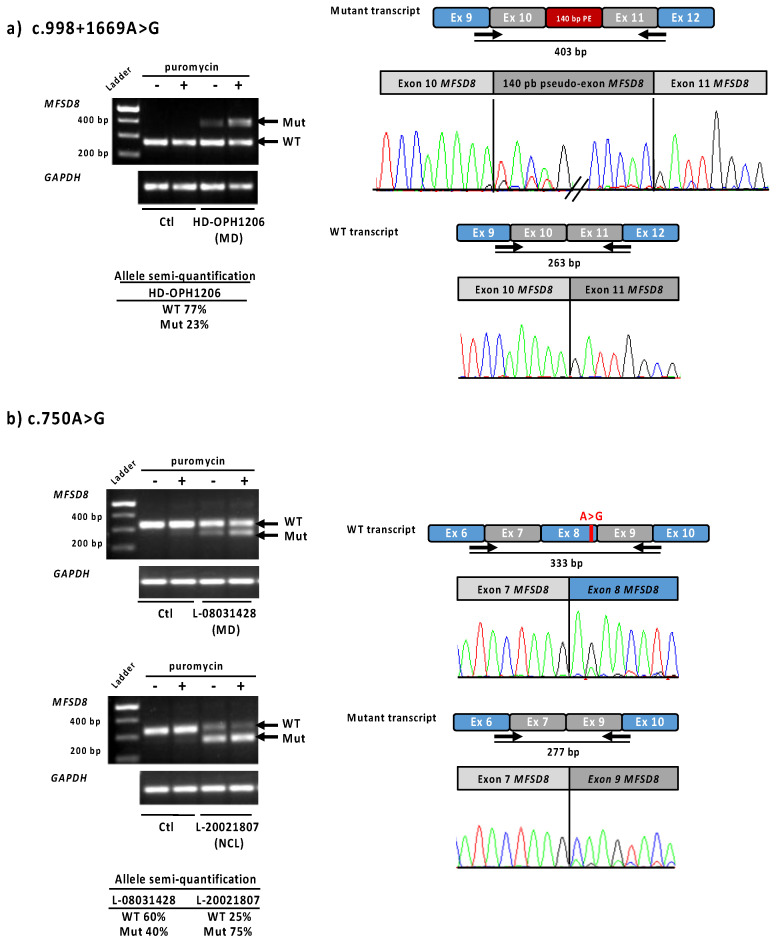
Functional tests results for *MFSD8* variants c.998+1669A>G and c.750A>G in LCLs Lymphoblastic cell lines from patients HD-OPH1206, L-08031428 and L-20021807 were used to analyse *MFSD8* cDNA expression. (**a**) Overlapping primers located at the exons 9–10 and 11–12, respectively were used to show the inclusion of a 140 bp pseudoexon due to the c.998+1669A>G variant. On the right, electropherograms obtained from the upper and lower bands separated on agarose gel. (**b**) The exon 8 skipping due to c.750A>G variant was confirmed using primers located at exons 6–7 and 9–10 junctions. The exon 8 skipping is observed in control at a very low level, suggesting exon 8 as an alternative cassette, normally expressed, and enhanced by the presence of this variant. Semi-quantification of the bands (using primers located in exon 7 and 9) has been performed on cells not treated by puromycin. Ctl—control; MD—macular dystrophy; NCL—neuronal ceroid lipofushinose.

**Table 1 ijms-23-04294-t001:** Clinical findings.

Patient ID, Sex	Age at Initial Examination	Symptoms	BCVA at First Examination	Refraction	Fundus	Colour Vision	Visual Field	ERG	SW-FAF	SD-OCT	Retinal Phenotype
**Syndromic IRD in Late-Infantile Ceroid Lipofuscinosis**											
L-20021807, M	10	VA loss and photophobia	20/400 OU	Cyclopleged +1.50 OU	Waxy pallor of ONH Severe vascular attenuation Yellowish macula Cellophane light reflex	NA	20° central scotoma at III3e target V4e normal	Unrecordable	Large area of increased autofluorescence with indistinct border in the posterior pole and the second more narrow annulus of increased autofluorescence around the fovea	Overall retinal thinning; Indistinct lamination and disappearance of outer layers (EZ and RPE)	EOSRD
**Isolated IRD**											
**Teenage-onset forms**											
L-08031428, M	12	VA loss, poor colour discrimination and photophobia	20/125 OU	Cyclopleged +1.25 (−0.50) 155° RE +1.75 (−0.75) 30° LE	Temporal pallor of ONH Arteriolar narrowing Punched-out round foveal lesion with hyperpigmented edges	Severe without-axis dyschromatopsia on Lantony D-15	Small relative central scotoma on static VF; Kinetic VF normal to all target sizes	Cone–rod dysfunction Reduced b/a ratio in DA 3.0 ERG	Round foveal hypoAF lesion	Foveal outer layer cavitation	CORD
IM-190703, M	12	VA loss, reading difficulties	20/32 OU	(−0.75) 90° RE −0.75 (−0.5) 90° LE	Temporal pallor of ONH Punched-out round foveal lesion with hyperpigmented edges	NA	Central scotoma	Scotopic normal Photopic reduced to 1/3 normal limits	Round hypoAF lesion	Loss of foveal outer reflective layers (ONL, EZ and RPE)	COD
VV-1595021, F	14	VA loss, reading difficulties, photophobia	20/40 OU	+0.75 OU	Punched out round foveal lesion with hyperpigmented edges	Ishihara normal	10° relative central scotoma at I1e target on static VF	Scotopic/photopic within normal limits	Round hypoAF lesion	Foveal outer layer cavitation	MD
**Adult-onset forms**											
HD-OPH1206, M	30	VA loss, photophobia (especially outdoors), contrast issues	20/63 RE 20/80 LE	−0.25 (−0.50) 170° RE −0.50 (−0.25) 80° LE	Punched-out round macular lesion with hyperpigmented edges	Severe without-axis dyschromatopsia on Lantony D-15	5° relative central scotoma at I1e target on static VF	Scotopic within normal limits Photopic reduced	Round hypoAF lesion with watershade edges	Loss of macular outer reflective layers (ONL, EZ and RPE)	COD
HD-OPH4200, F	37	VA loss, reading difficulties, asthenopia, reduction in *contrast* sensitivity	20/50 RE 20/32 LE	−1.75 (−0.25) 177° RE −2.00 (−0.25) 159° LE	Temporal pallor of ONH Loss of foveal light reflex	Red–green dyschromatopsia with the Farnsworth 100 Hue	5° relative central scotoma at I1e target on static VF	Scotopic within normal limits Photopic reduced	Round hypoAF lesion, surrounded by hyperAF edges	Loss of macular outer reflective layers (ONL, EZ and RPE)	COD
VV-51717, M	55	VA loss, reading difficulties, photophobia	20/100 OU	−0.50 (−0.50) 0° RE −0.75 (−0.50) 0° LE	Temporal pallor of ONH, Punched-out round macular lesion with hyperpigmented edges	Ishihara normal	NA	Scotopic/photopic within normal limits	Round hypoAF lesion, surrounded by hyperAF edges	Loss of macular outer reflective layers (ONL, EZ and RPE)	MD

BCVA, best corrected visual acuity; NTR, nothing to report; NA, unavailable; ONH, optic nerve head; RE, right eye; LE, left eye; OU, oculus utriusque (both eyes); VF, visual field; AF, autofluorescence; COD, cone dystrophy; CORD, cone–rod dystrophy; MD, macular dystrophy; LINCL, late-infantile neuronal ceroid lipofuscinosis; EOSRD, early onset severe retinal dystrophy.

**Table 2 ijms-23-04294-t002:** *MFSD8* (NM_152778.3) genotypes identified in our cohort.

Patient ID	Phenotype	Genomics Position (Hg19)	DNA Variant	Protein Variant	Variant Type	gnomAD AF	*In Silico* Prediction	Grantham Distance	ACMG Classification	Reference/ Phenotype
**Syndromic IRD**										
L-20021807	LINCL/ EOSRD	Chr4:128859942	**c.750A>G**	**p.[Arg233Serfs*5,=]**	Synonymous/Splicing	0	Exon 8 skipping SpliceAI: MFSD8|0.00|0.15|0.00|0.45|-4|49|0|-4	-	Likely Pathogenic (PS3,PM3,PP3,PP5)	Reith 2022 LINCL
		Chr4:128849857–128856974	**c.(755-2726_998+1981)delinsGTA**	**p.(Ser253Leufs*79)**	Copy Number Variation	0	Multiexons deletion (ex9-10)	-	Pathogenic (PVS1,PS3,PM2)	This study
**Isolated IRD**										
L-08031428	CORD	Chr4:128859942	**c.750A>G**	**p.[Arg233Serfs*5,=]**	Synonymous/Splicing	0	Exon 8 skipping SpliceAI: MFSD8|0.00|0.15|0.00|0.45|-4|49|0|-4	-	Likely Pathogenic (PS3,PM3,PP3,PP5)	Reith 2022 LINCL
		Chr4:128843111	c.1006G>A	p.(Glu336Lys)	Missense	0.00000882	CADD: 29.9	56	Likely Pathogenic (PM2,PM3,PM5,PP3)	This study
IM-190703	COD	Chr4:128878706 Chr4:128871002	c.104G>A* c.155G>C*	p.(Arg35Gln) p.(Gly52Ala)	Missense Missense	0.00000797 0	CADD: 28.0 CADD: 24,9	43 60	Unknown significance (PM2,PM3,PP3) Likely Pathogenic ((PM2,PM3,PM5,PP3)	This study
										This study
		Chr4:128842764	**c.1265C>A**	**p.(Ser422*)**	Nonsense	0	CADD: 37.0	-	Pathogenic (PVS1,PM2)	This study
VV-1595021	MD	Chr4:128843108	c.1009C>T	p.(Arg337Cys)	Missense	0.00001195	CADD: 28.6	180	Likely Pathogenic (PM2-S,PM3,PP3)	This study
		Chr4:128842888	**c.1141G>T**	**p.(Glu381*)**	Nonsense	0.000007075	CADD: 41.0	-	Pathogenic (PVS1,PM2,PP5-S)	Roosing 2015 MD
HD-OPH1206	COD	Chr4:128851907	c.929G>A	p.(Gly310Asp)	Missense	0	CADD: 23.9	94	Pathogenic (PP5-S,PM2,PP3)	Siintola 2007 LINCL
		Chr4:128850169	**c.998+1669A>G**	**p.[=,Lys333Asnfs*18]**	Deep Intronic Variation	0	140 bp pseudoexon insertion, SpliceAI: MFSD8|0.30|0.00|0.68|0.13|140|-3|1|-3	-	Likely Pathogenic (PS3,PM3,PM2,PP3)	This study
HD-OPH4200	COD	Chr4:128851955	**c.881C>A**	**p.(Thr294Lys)**	Missense	0.00000884	CADD: 24.4	78	Likely Pathogenic (PP5-VS,PM2-S,PP3)	Aiello 2009 LINCL
		Chr4:128843111	c.1006G>C	p.(Glu336Gln)	Missense	0.00301	CADD: 24.2	29	Likely Pathogenic (PS4,PP5-M,PP3,BS1)	Roosing 2015 MD
VV-51717	MD	Chr4:128843111	c.1006G>C	p.(Glu336Gln)	Missense	0.002494	CADD: 24.2	29	Likely Pathogenic (PS4,PP5-M,PP3,BS1)	Roosing 2015 MD
		Chr4:128851955	**c.881C>A**	**p.(Thr294Lys)**	Missense	0.000008018	CADD: 24.4	78	Likely Pathogenic (PP5-VS,PM2-S,PP3)	Aiello 2009 LINCL

AF, allele frequency; ACMG, American College of Medical Genetics classification [28]; CADD, Combined Annotation Dependent Depletion [29]; COD, cone dystrophy; CORD, cone–rod dystrophy; MD, macular dystrophy; LINCL, late-infantile neuronal ceroid lipofuscinosis; EOSRD, early onset severe retinal dystrophy; SpliceAI provides a score for acceptor gain, acceptor loss, donor gain and donor loss as well as the predicted position of the change [30]; Grantham distance assesses the physicochemical differences between two amino acid residues [31]; SNV, single-nucleotide variant class; * Variants c.104G>A and c.155G>C are in *cis* and form the complex allele c.[104G>A;155G>C]; In bold are the variants with a severe effect (based on Steenhuis et al., 2012, on in cellulo functional assay in this study, and on in silico missense analysis).

**Table 3 ijms-23-04294-t003:** *MFSD8* variants found in both MD and NCL patients from literature analysis.

Isolated Retinal Degeneration					Late-Infantile Neuronal Ceroid Lipofuscinosis		
Allele 1	Predicted Effect	Allele 2	Predicted Effect	Reference	Allele 2	Predicted Effect	Reference
c.750A>G p.[Arg233Serfs*5,=]	severe (exon skipping)	c.1006G>A p.(Glu336Lys)	mild (conservative substitution)	This study	c.(755-2726_998+1981)delinsGTA p.(Ser253Leufs*79)	severe (truncating)	This study
					c.750A>G p.[Arg233Serfs*5,=]	severe (exon skipping)	[27]
c.881C>A p.(Thr294Lys)	severe (enhanced proteolytic cleavage) [24]	c.1006G>C p.(Glu336Gln)	mild (conservative substitution)	This study (2 cases)	c.881C>A p.(Thr294Lys)	severe (enhanced proteolytic cleavage) [24]	[55] [56] (18 cases) [57]
					c.754+2T>A p.(?)	severe (splice variant)	[58]
c.929G>A p.(Gly310Glu)	severe (moderately conservative substitution)	c.998+1669A>G p.[=,Lys333Asnfs*18]	moderate (wt transcript 50%)	This study	c.863+3_4insT p.(?)	severe (splice variant)	[55]
					c.929G>A p.(Gly310Glu)	severe (moderately conservative substitution)	[50]
c.1006G>C p.(Glu336Gln)	mild (conservative substitution)	c.1141G>T p.(Glu381*)	severe (truncating)	[2]			
		c.1102G>C p.(Lys333Lysfs*3)	severe (truncating)	[2]			
		c.103C>T p.(Arg35*)	severe (truncating)	[13]			
		c.1394G>A p.(Arg465Gln)	supposed severe (although conservative substitution)	[13]			
		c.233G>A p.(Trp78*)	severe (truncating)	[13]			
		c.881C>A p.(Thr294Lys)	severe (enhanced proteolytic cleavage) [24]	This study (2 cases)			
c.1066C>T p.(Pro356Ser)	moderate (moderately conservative substitution)	c.1102+2T>C p.(?)	severe (splice variant)	[17]			
c.1141G>T p.(Glu381*)	severe (truncating)	c.1006G>C p.(Glu336Gln)	mild (conservative substitution)	[2]			
		c.1009C>T p.(Arg337Cys)	moderate (radical substitution)	This study			
c.1235C>T p.(Pro412Leu)	severe (increased proteolytic cleavage) [24]	c.1361T>C p.(Met454Thr)	moderate (moderately conservative substitution)	[13] (6 families) [15]	c.1235C>T p.(Pro412Leu)	severe (increased proteolytic cleavage) [24]	[59]
c.1265C>A p.(Ser422*)	severe (truncating)	c.[104G>A;155G>C] p. [(Arg35Gln;Gly52Ala]	mild (conservative substitution)	This study			
c.1361T>C p.(Met454Thr)	moderate (moderately conservative substitution)	c.1361T>C p.(Met454Thr)	moderate (moderately conservative substitution)	[13] (6 families) [15]	in cis with c.1219T>C p.(Trp407Arg) at the homozygous state	severe (moderately radical)	[60]
		c.1235C>T p.(Pro412Leu)	severe (increased proteolytic cleavage) [24]	[15]			
c.1394G>A p.(Arg465Gln)	supposed severe (although conservative substitution)	c.1006G>C p.(Glu336Gln)	mild (conservative substitution)	[13]	c.1394G>A p.(Arg465Gln)	supposed severe (although conservative substitution)	[56]

In grey background are the variants with a severe effect (based on Steenhuis et al., 2012, on in cellulo functional assay in this study, and on in silico missense analysis). Allele 1 corresponds to variants identified in MD cases and in NCL cases. Allele 2 is the variant found in trans, either in MD or in NCL. Predicted impact of the variants are presented. ND, not described. Arg465Gln substitution is conservative (Grantham score 43) but Arg465 is a positively charged polar residue, interacting with Gln87 in the second transmembrane domain through a hydrogen bond. Replacement by Gln, an uncharged residue, could destabilize the protein conformation. We therefore assigned it as severe. Of note, in Patiño et al. 2014, p.(Met454Thr) was in *cis* with a severe variant p.(Thr407Arg) at the homozygous state, reinforcing the moderate effect of p.(Met454Thr).

## Data Availability

All data are contained within the article or Appendix A.

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
