# Peer review of "Contribution of Whole-Genome Sequencing and Transcript Analysis to Decipher Retinal Diseases Associated with MFSD8 Variants"

_ijms, 2022, doi:10.3390/ijms23084294_

Round 1

Reviewer 1 Report

Interesting study. Pictures are informative. Some of them have follow-up Slit lamp pictures and OCT, some don’t. Please add follow-ups for all, if available.

Another minor comment:

Line 217-218: “One patient with MD (L-08031428) and one with LINCL (L-20021807), both native from the North of France, carried the same variant c.750A>G, not reported in gnomAD.”

Please mention the age.

Author Response

Interesting study. Pictures are informative. Some of them have follow-up Slit lamp pictures and OCT, some don’t. Please add follow-ups for all, if available.

We are grateful for the reviewer's positive review and valuable suggestions.

Unfortunately, we have follow-up only for the oldest patients. The remaining ones are young (10 and 12 years old) and have been examined only recently, explaining the absence of modification in their clinical presentation. We addressed this point in the text and added a follow-up picture in Figure 1.

Line 217-218: “One patient with MD (L-08031428) and one with LINCL (L-20021807), both native from the North of France, carried the same variant c.750A>G, not reported in gnomAD.” Please mention the age.

We added ages for both patients in the text.

Reviewer 2 Report

The authors report the identification of 7 patients with biallelic variants in MFSD8, 6 of which with retinal dystrophies and 1 with neuronal ceroid lipofuscinosis, among a cohort of 1185 patients with retinal dystrophies. This report confirms that MFSD8 can cause isolated retinal dystrophy, besides neuronal ceroid lipofuscinosis. The work also shows how the integrated analysis of DNA and RNA can be useful to identify elusive variants. Nevertheless, the manuscript is unclear in describing the overall experimental design and the diagnostic tests pipeline, making it difficult to appreciate its scientific relevance.

In this reviewer’s opinion, a careful evaluation of the manuscript can be performed after that the authors have addressed these points:

  • It is not clear how many patients have been discovered with mutations in MFSD8. The manuscript results section reports 7 patients, abstract states 7 (French) + 2 (Swiss), the supplementary Figure S3 states 5. Please, revise and make them coherent.
  • The initial study cohort is unclear. The abstract mentions two initial cohorts 3348 and 1185 of French patients. I understood that only the 1185 cohort underwent some genetic panel that included MFSD8. Why was the genetic test not conducted on the other 3348-1185=2163 patients? How were those cohorts composed?
  • Two patients came from a Swiss cohort. Please, indicate them among those in the results. Why have these patients been added? Do they contribute to illustrating MFSD8 molecular or clinical features?
  • Please better clarify the genetic tests that have been performed on the patients. All the 1185 patients have been sequenced for the MFSD8 gene? How many and who has been whole-genome sequenced? How many and who has been whole-gene MFSD8 sequenced? How many and who has been RNA-studied? How these patients have been selected?
  • For the reported variants, please state the ACMG criteria that bring to pathogenicity classification.

Author Response

The authors report the identification of 7 patients with biallelic variants in MFSD8, 6 of which with retinal dystrophies and 1 with neuronal ceroid lipofuscinosis, among a cohort of 1185 patients with retinal dystrophies. This report confirms that MFSD8 can cause isolated retinal dystrophy, besides neuronal ceroid lipofuscinosis. The work also shows how the integrated analysis of DNA and RNA can be useful to identify elusive variants. Nevertheless, the manuscript is unclear in describing the overall experimental design and the diagnostic tests pipeline, making it difficult to appreciate its scientific relevance.

In this reviewer’s opinion, a careful evaluation of the manuscript can be performed after that the authors have addressed these points:

We sincerely thank the reviewer for his/her comments and valuable suggestions.

We agree that the manuscript should benefit from a more detailed description of the cohort. We therefore addressed these comments in the text and response point by point below. Figure S3 (now Figure S2) was reassessed as well and the total number of patients in each group was recalculated.

It is not clear how many patients have been discovered with mutations in MFSD8. The manuscript results section reports 7 patients, abstract states 7 (French) + 2 (Swiss), the supplementary Figure S3 states 5. Please, revise and make them coherent.

The initial study cohort is unclear. The abstract mentions two initial cohorts 3348 and 1185 of French patients. I understood that only the 1185 cohort underwent some genetic panel that included MFSD8. Why was the genetic test not conducted on the other 3348-1185=2163 patients? How were those cohorts composed?

The Lille cohort contains 3,348 probands, all of them have been genetically assessed since 2014. These patients were genotyped firstly using a NGS panel of the 18 most frequent genes assessed in our laboratory(ABCA4, PRPH2, ELOVL4, CRX, GUCY2D, BEST1, IMPG1, IMPG2, RHO, PRPF31, RP1, PRPF3, PRPF6, RPGR, RP2, NRL, IMPDH1, NR2E3). Then, this small panel was replaced by a large NGS panel containing 156 and thereafter 230 IRD genes, both containing MFSD8 gene. The total number of patients being analyzed by this large panel is 1,049.

Among them, 340 patients had a clinical diagnosis of cone dystrophy, cone-rod dystrophy and macular dystrophy including Stargardt disease. The 5 cases harbouring bi-allelic MFSD8 variants were found in this last group. No bi-allelic variants in MFSD8 was found in the 709 “other IRD” patients.

We agree that we cannot exclude the presence of other patients harbouring MFSD8 variants, among the 356 unsolved patients with CD or MD, analyzed by a small panel. A complementary study should be conducted to address this point. We added this comment in the text.       

We renamed and clarify the Figure S3 (now Figure S2): "Molecular diagnosis flowchart in the Lille cohort”. This figure illustrates the 5 cases identified in the Lille laboratory, but doesn't include the 2 swiss patients.

The following legend to Figure S3 was added: "The 3,348 probands of the Lille cohort underwent a genetic analysis, either by next generation sequencing (NGS) of a large gene panel including MFSD8 gene (n=1,049), or by NGS of a small panel of targeted genes not including MFSD8 (n=2,299). Each category is divided into two groups: a group named "other IRDs" with patients presenting Rod-Cone Dystrophy, Leber Congenital Amaurosis, Congenital Stationary Night Blindness, Choroideremia (n= 709 analyzed by a large NGS panel; n= 1,346 analyzed by a small NGS panel); and a second group with patients presenting Cone Dystrophy, Macular Dystrophy including Stargardt disease (n= 340 analyzed by a large NGS panel; n= 953 analyzed by a small NGS panel. ABCA4 is the most frequent gene identified in this group with 549 STGD1 cases identified by the small panel and 48 by the large panel (n=597). The presence of other patients harbouring MFSD8 variants among the 356 unsolved patients with CD or MD, analyzed by a small panel cannot be ruled out. The two swiss patients described in this study were not part of this initial cohort. "

The abstract was also modified.

Two patients came from a Swiss cohort. Please, indicate them among those in the results. Why have these patients been added? Do they contribute to illustrating MFSD8 molecular or clinical features?

Indeed, the two swiss patients have not been diagnosed in the Lille Center and are therefore not part of the initial Cohort. They have been added to illustrate MFSD8 molecular and clinical features, as this gene is rarely associated with macular dystrophy. Their insertion in the manuscript strengthens the clinical description and the assessment of the pathogenicity of the MFSD8 variants found.

Please better clarify the genetic tests that have been performed on the patients. All the 1185 patients have been sequenced for the MFSD8 gene? How many and who has been whole-genome sequenced? How many and who has been whole-gene MFSD8 sequenced? How many and who has been RNA-studied? How these patients have been selected?

As mentioned above, only 340 patients presenting CD, CRD or MD have been tested for MFSD8 using a large NGS panel. We cannot exclude that some of the other unsolved cases with CD, CRD or MD, that did not benefit from a large panel (n =356), could carry MFSD8 variants.

WGS has been performed for patient HD-OPH1206, as we searched for a second intronic variant in MFSD8, as well as for patient L-20021807, to delineate the border of the large deletion detected by NGS panel.

To prove the pathogenicity of the novel splice variants (c.998+1669A>G and c.750G>A) at the RNA level, we requested novel samples for the 3 probands (and their family when available) carrying those splice variants. From these samples, lymphocytes were immortalized, treated or not by puromycin, a nonsense mediated decay inhibitor, and RNA was then extracted to assess their splice effect.

For the reported variants, please state the ACMG criteria that bring to pathogenicity classification.

ACMG criteria have been added in Table 2 for all variants.